# Fingerprints of hot-phonon physics in time-resolved correlated quantum lattice dynamics

E. Cappelluti[1*], D. Novko[2,3]

**1** Istituto di Struttura della Materia, CNR (ISM-CNR), 34149 Trieste, Italy
**2** Institute of Physics, 10000 Zagreb, Croatia
**3** Donostia International Physics Center (DIPC), 20018 Donostia-San Sebastián, Spain
\* emmanuele.cappelluti@ism.cnr.it

July 28, 2021

## Abstract

The time dynamics of the energy flow from electronic to lattice degrees of freedom in pump-probe setups could be strongly affected by the presence of a hot-phonon bottleneck, which can sustain longer coherence of the optically excited electronic states. Recently, hot-phonon physics has been experimentally observed and theoretically described in $MgB_2$, the electron-phonon based superconductor with $T_c \approx 39$ K. By employing a combined ab-initio and quantum-field-theory approach and by taking $MgB_2$ as an example, here we propose a novel path for revealing the presence and characterizing the properties of hot phonons through a direct analysis of the information encoded in the lattice inter-atomic correlations. Such method exploits the underlying symmetry of the $E_{2g}$ hot modes characterized by a out-of-phase in-plane motion of the two boron atoms. Since hot phonons occur typically at high-symmetry points of the Brillouin zone, with specific symmetries of the lattice displacements, the present analysis is quite general and it could aid in revealing the hot-phonon physics in other promising materials, such as graphene, boron nitride, or black phosphorus.

# 1 Introduction

Ultrafast time-resolved measurements have proven in the past few decades to be a powerful tool for investigating fundamental mechanisms of a variety of physical phenomena and for revealing processes governing different states of the matter, where several degrees of freedom (e.g., electrons, lattice and magnetic modes) play a mutual role [1–13]. In a typical pump-probe setup, energy is initially injected into the electronic degrees of freedom by means of particle-hole excitations, giving rise to a non-thermal electronic distribution. The time evolution of such non-thermal excitations is governed by different scattering mechanisms that lead to an energy transfer towards other degrees of freedom, in particular in the lattice mode sector, until, on a long time scale, a new final steady state is reached where all the different degrees of freedom are thermalized and at equilibrium.

The possibility of sustaining non-thermal states for sufficiently long time opens striking opportunities in the field of quantum information as long as the scattering sources affecting such states can be kept under control. The electron-phonon (el-ph) coupling is of major importance in this context since it represents one of the primary channels responsible for the internal thermalization of the electronic degrees of freedom, and because it leads eventually, when it is not constrained, to a system heating detrimental for any coherent process [14].

However, the thermalization process of some materials can be hindered by the so-called phonon bottleneck [2, 3, 15–34] that quenches the energy transfer between electron and lattice degrees of freedom. Such a physical phenomenon is usually encountered in semiconductors and semimetals where the pump-driven particle-hole excitations are restricted to few single points (valleys) in the Brillouin zone. In such conditions only a few phonons with the right symmetry and with selected momentum $\mathbf{q}$ connecting two valleys can be excited efficiently. Energy from the electron sub-system is thus prevalently transferred to these modes that get "hot", meaning they acquire a phonon population $n_{\mathbf{q}}$ much larger than other modes, giving rise to a non-thermal phonon distribution.

Within this context, the possibility of a phonon bottleneck, and hence of hot phonons, has been so far associated only to semiconductors and semimetals, whereas conventional metals, with a large Fermi surface allowing scattering with many $\mathbf{q}$-phonons, were thought to be incompatible with such a scenario. At odds with this belief, a novel path for inducing hot phonons has been recently proposed for unconventional metals characterized by a strong anisotropy of the el-ph coupling [35, 36]. In the paradigmatic case of $MgB_2$, the el-ph coupling is concentrated in a few in-plane modes at the Brillouin zone center possessing the $E_{2g}$ symmetry [37–44], with a strong resemblance to the relevant modes in single- and multi-layer graphene. As a consequence of such remarkable anisotropy, the energy initially pumped into the electron sector is efficiently and rapidly ($\sim 50$ fs) transferred only to such few lattice degrees of freedom that get a much larger population compared to the remaining lattice modes, whereas a final thermalization among the whole phonon modes and among electron and lattice degrees of freedom occurs on a much larger time scale ($\sim 0.4$ ps) [36]. On the experimental ground, an indirect signature of such hot-phonon scenario has been observed in the onset of unconventional features in the time-resolved optical spectra [35]. More direct and precise fingerprints have been proposed at the theoretical level in the analysis of time-resolved Raman spectroscopy, both in spectral features as well as in integrated spectral weights [36]. An accurate measurement of time-resolved Raman spectra in $MgB_2$ needs to face with the limitations of the uncertainty principle constraining time and energy resolution [45]. In addition, it should be remarked that in both cases the presence of a non-thermal population of the in-plane $E_{2g}$ modes is not directly probed via the properties of the lattice degrees of freedom, but indirectly through the el-ph driven

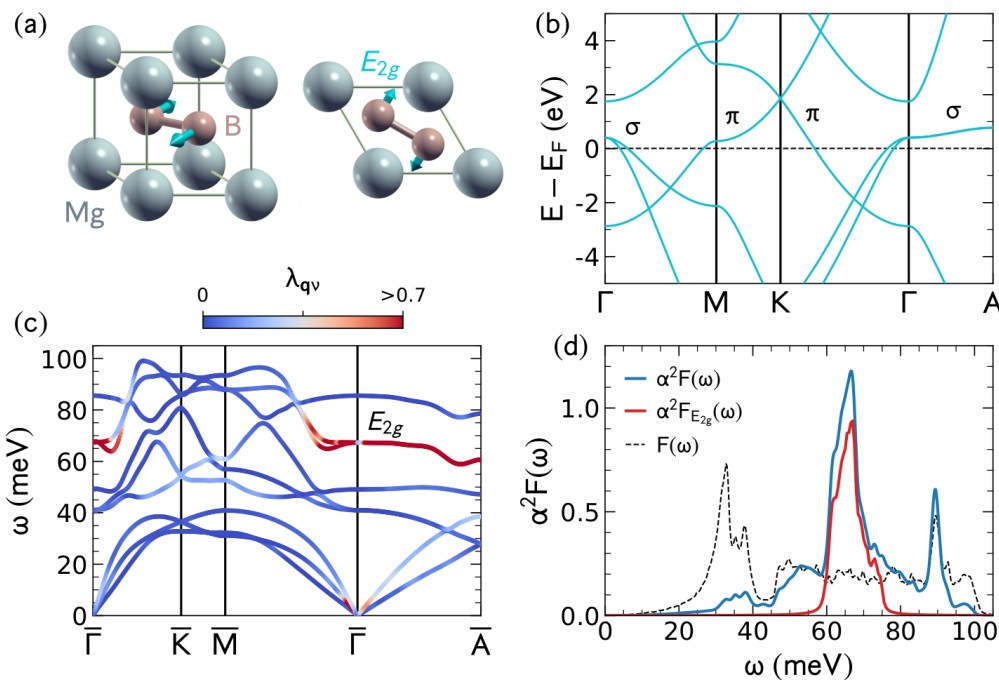

Figure 1: (a) Crystal structure of $MgB_2$. Also displayed are the lattice displacements of the in-plane $E_{2g}$ mode at $\mathbf{q} = 0$. (b) Electronic band dispersion of $MgB_2$. Bands are labelled as $\sigma$- and $\pi$-bands according to their symmetry. (c) Phonon dispersions of $MgB_2$. The color code (color bar atop) represents the strength of the el-ph coupling $\lambda_{\mathbf{q}\nu}$. (d) Corresponding phonon density of states $F(\omega)$ (dashed line) and the total Eliashberg function $\alpha^2 F(\omega)$ (blue solid line). The solid red line shows the contribution to the Eliashberg function associated with the hot $E_{2g}$ modes.

many-body renormalization of the electronic response, i.e. in the optical response [35] or in the phonon self-energy [36,46], which is strictly related to the electronic screening.

In the present work we suggest that a smoking gun evidence of the presence of hot phonons in $MgB_2$ can be attained in a more straightforward way by the analysis of the time-resolved lattice dynamics, as obtained by means of the X-ray or neutron scattering, for instance. In particular, we show that the inter-atomic pair-distribution function, which encodes information about the correlated atomic motion, is highly sensitive to the presence of hot-phonon and it provides a striking tool for detecting them. As related to the analysis of time-independent quantities, our proposal is not affected by the limitations of uncertainty principle. Note that the present analysis, here applied to the paradigmatic case of $MgB_2$, does not rely on specific properties of this material but it can be employed as well in other compounds, such as single-layer and multi-layer graphene [47,48], transition metal dichalcogenides [49], black phosphorous [34,50], and hole-doped diamond [51,52], where el-ph coupling plays a major role.

## 2  Theoretical modelling

The crystal structure of $MgB_2$ is rather simple, with hexagonal graphene-like planes of B atoms spaced vertically by Mg atoms located in the center of the hexagons (see Fig. 1a) [53]. In order to reproduce an *ab-initio* ground state properties of this material,

we employ here the QUANTUM ESPRESSO package [54]. Norm-conserving pseudopotentials were employed with the Perdew-Burke-Ernzerhof exchange-correlation functional [55]. The in-plane lattice parameter and the distance between two boron planes were set to 3.083 Å and 3.521 Å, respectively. A $24 \times 24 \times 24$ Monkhorst-Pack grid in momentum space and a plane-wave cutoff energy of 60 Ry were used for ground-state calculations. The phonon dispersion was calculated on a $12 \times 12 \times 12$ grid using density functional perturbation theory [56], and the electron-phonon coupling was computed by using the EPW code [57]. Electron energies, phonon frequencies, and electron-phonon coupling matrix elements were interpolated using maximally-localized Wannier functions [58]. The Eliashberg function, which shows a strength of the el-ph coupling for a particular phonon energy, was obtained on a $40 \times 40 \times 40$ grid of electron and phonon momenta.

The well-known band structure is shown in Fig. 1b. Due to the symmetry of the systems, the in-plane $\sigma$-bands retain a strong two-dimensional character with a very weak amount of Mg, very similar to the $\sigma$ bands of graphene, whereas the $p_z$ B-orbitals strongly hybridize with Mg giving rise to $\pi$-bands with a strong three-dimensional character. As a consequence of such strong anisotropy in the electronic degrees of freedom, the el-ph coupling results to be highly anisotropic, as depicted in Fig. 1c. The el-ph coupling among states in the $\pi$ bands and among $\sigma$ and $\pi$ states, involving $\pi$ bands with a good metallic character, is quite weak. On the other hand, the $\sigma$ bands appear to be only slightly doped, resulting in a poor screening of the in-plane B-B lattice modes, and in a corresponding large el-ph coupling between states in the $\sigma$ bands. A further consequence of the small hole-doping of the two-dimensional $\sigma$ bands is the intrinsic restriction, obeying to the Lindhard model, of a sizable el-ph coupling in a small momentum region $|\mathbf{q}_{||}| \leq 2k_{\mathrm{F}}$, [42,59–62] where $\mathbf{q}_{||} = (q_x, q_y)$ and $k_{\mathrm{F}}$ is the in-plane Fermi momentum of the $\sigma$ bands which is only weakly dependent on $k_z$.

Within the above framework, a predominant role is played by the in-plane $E_{2g}$ phonon modes at small $\mathbf{q}_{||}$, characterized by a strong el-ph coupling and which corresponds to out-of-phase displacements of the two boron atoms within the unit cell, as depicted in Fig. 1a. The overall strongly-coupled modes are thus limited, in a schematic way, to only two phonon branches within the energy window $\omega_{\mathbf{q},\nu} \in [60:75]$ meV, with roughly $|\mathbf{q}_{||}| \leq 2k_{\mathrm{F}}$ and weak dispersion along $q_z$, amounting roughly to 5 % of the total phonon modes, and corresponding thus to a very small specific heat capacity. The crucial role of the few $E_{2g}$ modes in the total coupling can be clearly pointed out in the Eliashberg function (Fig. 1d) which shows a remarkable peak in the range $\omega_{\mathbf{q},\nu} \in [60:75]$ meV, in spite of the absence of any particular feature in the corresponding phonon density of states.

Following Ref. [36], the predominance of the hot-phonon modes in the total coupling can be captured by splitting the total Eliashberg function in a *hot* and *cold* component, $\alpha^2 F(\omega) = \alpha^2 F_{\mathrm{hot}}(\omega) + \alpha^2 F_{\mathrm{cold}}(\omega)$, where $\alpha^2 F_{\mathrm{hot}}(\omega)$ contains the contribution of the hot $E_{2g}$ modes close to the $\overline{\Gamma} - \overline{\mathrm{A}}$ path in the relevant energy range $\omega \in [60:75]$ meV (the phonon modes colored in red in Fig. 1c), while $\alpha^2 F_{\mathrm{cold}}(\omega)$ takes into account the other weakly coupled cold modes. A similar splitting can be performed for the phonon density of states $F(\omega) = F_{\mathrm{hot}}(\omega) + F_{\mathrm{cold}}(\omega)$.

Equipped with such first-principle input, we can describe the time-dynamics of the energy transfer between the different degrees of freedom in terms of three temperatures [1, 9, 36, 47, 48, 63–66], i.e., an electron one $T_{\mathrm{e}}$, a hot-phonon temperature $T_{\mathrm{hot}}$, which governs the population of the $E_{2g}$-like strongly-coupled hot-phonon modes, and the cold-lattice temperature $T_{\mathrm{cold}}$ that describes the temperature of the weakly coupled cold modes. The time evolution of the these characteristic temperatures upon a pump pulse can be

described by the set of coupled equations

$$C_{\mathrm{e}}\frac{\partial T_{\mathrm{e}}}{\partial t} = S(z,t) + \nabla_z(\kappa\nabla_z T_{\mathrm{e}}) - G_{\mathrm{hot}}(T_{\mathrm{e}} - T_{\mathrm{hot}}) - G_{\mathrm{cold}}(T_{\mathrm{e}} - T_{\mathrm{cold}}), \quad (1)$$

$$C_{\mathrm{hot}}\frac{\partial T_{\mathrm{hot}}}{\partial t} = G_{\mathrm{hot}}(T_{\mathrm{e}} - T_{\mathrm{hot}}) - C_{\mathrm{hot}}\frac{T_{\mathrm{hot}} - T_{\mathrm{cold}}}{\tau_0}, \quad (2)$$

$$C_{\mathrm{cold}}\frac{\partial T_{\mathrm{cold}}}{\partial t} = G_{\mathrm{cold}}(T_{\mathrm{e}} - T_{\mathrm{cold}}) + C_{\mathrm{hot}}\frac{T_{\mathrm{hot}} - T_{\mathrm{cold}}}{\tau_0}. \quad (3)$$

Here $C_{\mathrm{e}}$, $C_{\mathrm{hot}}$, and $C_{\mathrm{cold}}$ are the specific heat capacities for the electron, hot-phonon, and cold-phonon states, respectively, and $G_{\mathrm{hot}}/G_{\mathrm{cold}}$ are the electron-phonon relaxation rates between electronic states and hot/cold phonons modes. They can be computed as

$$C_{\mathrm{e}} = \int_{-\infty}^{\infty} d\varepsilon N(\varepsilon)\varepsilon\frac{\partial f(\varepsilon - \mu; T_{\mathrm{e}})}{\partial T_{\mathrm{e}}}, \quad (4)$$

$$C_{\mathrm{hot}} = \int_{0}^{\infty} d\omega F_{E_{\mathrm{hot}}}(\omega)\omega\frac{\partial b(\omega; T_{E_{2g}})}{\partial T_{E_{2g}}}, \quad (5)$$

$$C_{\mathrm{cold}} = \int_{0}^{\infty} d\omega F_{\mathrm{cold}}(\omega)\omega\frac{\partial b(\omega; T_{\mathrm{cold}})}{\partial T_{\mathrm{cold}}}, \quad (6)$$

$$G_{\mathrm{hot}} = \frac{2\pi k_B}{\hbar}N(\mu)\int d\Omega\Omega\alpha^2 F_{E_{2g}}(\Omega), \quad (7)$$

$$G_{\mathrm{cold}} = \frac{2\pi k_B}{\hbar}N(\mu)\int d\Omega\Omega\alpha^2 F_{\mathrm{cold}}(\Omega). \quad (8)$$

Here $N(\varepsilon)$ is the electronic density of states, $\mu$ the electronic chemical potential, and $f(x;T) = 1/[\exp(x/T) + 1]$, $b(x;T) = 1/[\exp(x/T) - 1]$ are the Fermi-Dirac and the Bose-Einstein distribution functions, respectively. The values obtained from the first-principles calculations are $C_{\mathrm{e}} = 90\,\mathrm{J/m^3 K^2} \times T_{\mathrm{e}}$, $C_{\mathrm{hot}} = 0.13\,\mathrm{J/m^3 K}$, and $C_{\mathrm{cold}} = 4.1\,\mathrm{J/m^3 K}$, $G_{\mathrm{hot}} = 2.8 \times 10^{18}\,\mathrm{W/m^3 K}$ and $G_{\mathrm{cold}} = 3.6 \times 10^{18}\,\mathrm{W/m^3 K}$. Furthermore, modelling a pump-probe setup, the term $S(z,t) = I(t)e^{-z/\delta}/\delta$ takes into account the energy absoption from the laser pulse into the electronic degrees of freedom, where $I(t)$ is the intensity of the absorbed fraction of the laser pulse (with a Gaussian profile) and $\delta$ is the penetration depth. Finally, the relaxation time $\tau_0$ takes into account the scattering between hot and cold modes driven by the anharmonic phonon-phonon coupling, for which we take an estimate of 400 fs [67].

The time evolution of the three characteristic temperatures $T_{\mathrm{e}}$, $T_{\mathrm{hot}}$, $T_{\mathrm{cold}}$ (Fig. 2a) shows a remarkable increase of the effective temperature of the hot modes $T_{\mathrm{hot}}$, which reaches a peak $T_{\mathrm{hot}}^{\max} \approx 1250$ K at $t^* \approx 50$ fs with a small delay compared to the time behavior of the electronic temperature $T_{\mathrm{e}}$. The remnant cold phonon modes follow, on the other hand, a completely different behavior with a slow monotonic increase towards the final equilibrium state at $t \geq 0.3 - 0.4$ ps where all the degrees of freedom are thermalized with each other.

Such a behavior, showing a clear hot-phonon scenario, can be compared with the isotropic case where the energy from the electronic degrees of freedom is transferred in a equal way to *all* the lattice modes, without a preferential channel, and one can thus define a unique phonon temperature $T_{\mathrm{ph}}$ for all the modes [63]. The coupled equations of the two temperature model can be obtained by setting $T_{\mathrm{hot}} = T_{\mathrm{cold}} \equiv T_{\mathrm{ph}}$ in Eqs. (1)-(3):

$$C_{\mathrm{e}}\frac{\partial T_{\mathrm{e}}}{\partial t} = S(z,t) + \nabla_z(\kappa\nabla_z T_{\mathrm{e}}) - G_{\mathrm{ph}}(T_{\mathrm{e}} - T_{\mathrm{ph}}), \quad (9)$$

$$C_{\mathrm{ph}}\frac{\partial T_{\mathrm{ph}}}{\partial t} = G_{\mathrm{ph}}(T_{\mathrm{e}} - T_{\mathrm{ph}}),, \quad (10)$$

where $C_{\mathrm{ph}} = C_{\mathrm{hot}} + C_{\mathrm{cold}}$ and $G_{\mathrm{ph}} = G_{\mathrm{hot}} + G_{\mathrm{cold}}$. The time evolution of $T_{\mathrm{e}}$, $T_{\mathrm{ph}}$ is displayed in Fig. 2b. In the absence of the hot-phonon bottleneck, a much faster thermalization

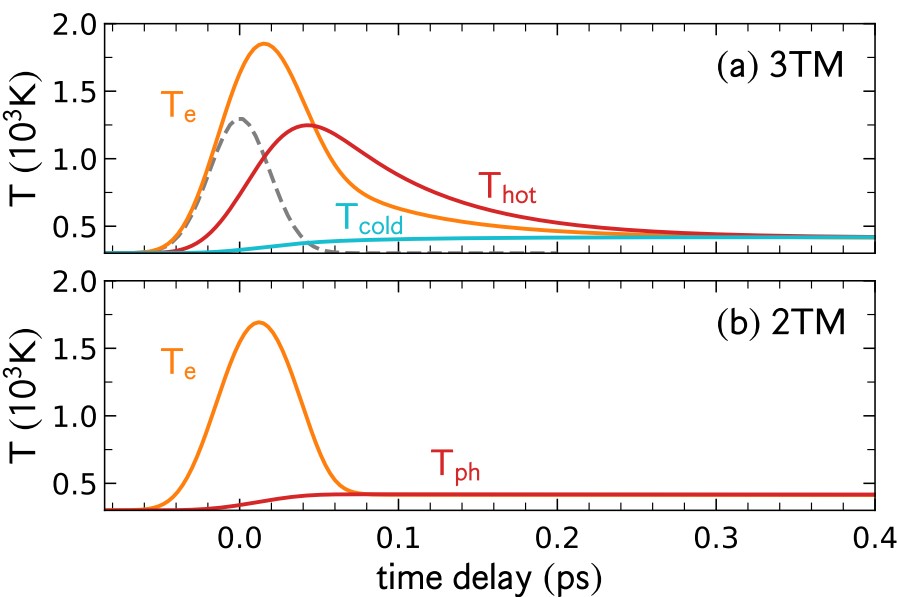

Figure 2: (a) Time evolution of the characteristic effective temperatures $T_e$, $T_{hot}$, $T_{cold}$ for the three-temperature model suitable for hot phonons. The grey dashed line shows the pulse profile, with the pulse duration of 45 fs and an absorbed fluence of $12\,\mathrm{J/m^2}$. (b) Similar as in panel (a) within the assumption of thermal distribution (two-temperature model) of the lattice modes as described with Eqs. (9) and (10).

between the electrons and the lattice degrees of freedom is obtained, within a time-scale of $t \sim 0.08$ ps, while the average phonon temperature $T_{ph}$ reaches the maximum of about 420 K.

## 3  Time-resolved lattice dynamics

As discussed above, the hot-phonon scenario is characterized by the regime $T_{hot} \gg T_{cold}$, where the population of the hot-lattice modes is singled out and is significantly larger than the population of the other cold modes. Equipped with the detailed input from the *ab initio* calculations, we discuss now how this scenario in MgB$_2$ can be revealed directly in the dynamical lattice properties, i.e., in the amplitude of the mean square lattice displacements, resolved for atomic species and for different directions; and more strikingly in the degree of *correlation* of the atomic motion. The peculiar properties of hot phonons in MgB$_2$ rely on the $E_{2g}$ symmetry of the strongly coupled hot modes, in particular: (*i*) the pure in-plane B character of the $E_{2g}$ modes; (*ii*) the counter-phase motion of the two B atoms per cell that move in opposite direction, as depicted in Fig. 1a.

The evaluation of the mean square lattice displacements based on the force-constant-model calculations has been detailed in Refs. [68,69]. Here, we extend such computation by using fully *ab initio* methodology and, more importantly, by introducing the hot-phonon scenario and specifying a different temperatures for each relevant modes. More explicitly, we can define the projected mean-square lattice displacement $\sigma^2(i_\alpha)$ as

$$\sigma^2(i_\alpha) = \langle [\mathbf{u}_i \cdot \hat{\mathbf{r}}_\alpha]^2 \rangle, \tag{11}$$

where $\mathbf{u}_i$ is the lattice displacement of atom $i$ from its average position and $\hat{\mathbf{r}}_\alpha$ is the unit

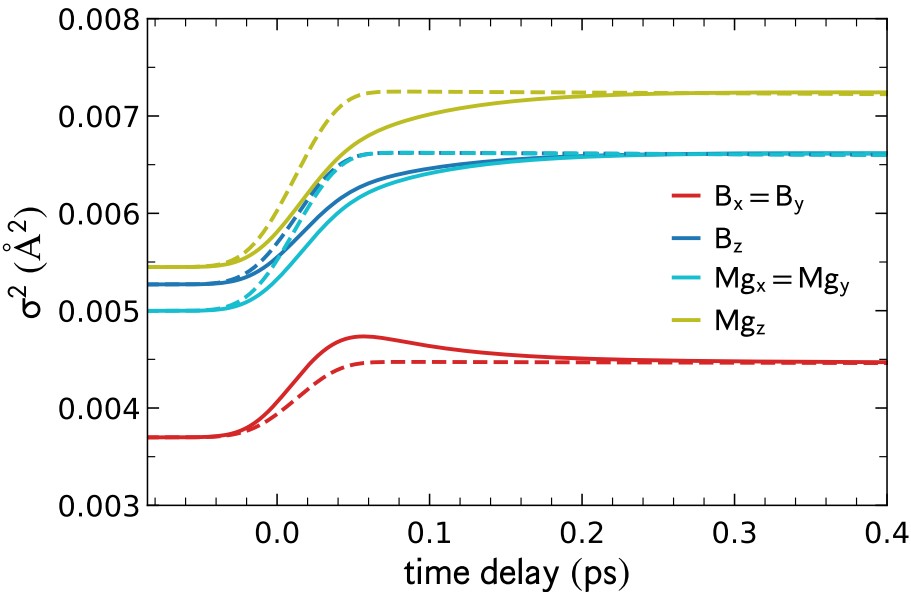

Figure 3: Time evolution of the mean square lattice displacements $\sigma^2$ for each atom and for different axis directions in the case of the hot-phonon scenario, described by Eqs. (1)-(3) (full lines). The corresponding results for the thermal phonon distribution [described by Eqs. (9)-(10)] are shown with the dashed lines.

vector pointing along the direction $\alpha = x, y, z$. On the microscopic ground we can write

$$\sigma^2(i_\alpha) = \frac{\hbar}{N} \sum_{\mathbf{q},\mu} \frac{|\epsilon_{\mathbf{q},\mu}^{i\alpha}|^2}{M_i \omega_{\mathbf{q},\mu}} \left[ \frac{1}{2} + n\left( \frac{\omega_{\mathbf{q},\mu}}{T_{\mathbf{q},\mu}} \right) \right], \tag{12}$$

where $N$ is the total number of $\mathbf{q}$-points considered in the phonon Brillouin zone, $M_i$ is the atomic mass of atom $i$, $\omega_{\mathbf{q},\mu}$ is the frequency of a phonon mode of branch $\mu$ with momentum q, $T_{\mathbf{q},\mu} = T_{\text{hot}}, T_{\text{cold}}$ is the appropriate temperature for such mode, and $\epsilon_{\mathbf{q},\mu}^{i\alpha}$ is the component of the corresponding eigenvector $\hat{\epsilon}_{\mathbf{q},\mu}$ describing the displacement of the $i$ atom along the $\alpha$ direction.

The behavior of the mean square lattice displacements resolved for each atom and for different axis directions in the case of hot non-thermal phonon distribution is shown in Fig. 3 (solid lines). As expected, compared with the case of a thermal phonon distribution described by Eqs. (9)-(10) (dashed lines), we do not see any remarkable difference for the phonon modes with Mg character and with out-of-plane lattice displacements, apart from a slower thermalization in the case of hot phonons, due to the longer storage of energy in the hot-phonon modes. A slight effect of the presence of hot phonons can be detected in the behavior of the mean square in-plane B lattice displacements which shows a non-monotonic behavior with a peak value larger than the final equilibrium state, at $t^* = 50\,\text{fs}$, in fair accordance with the peak of $T_{\text{hot}}$.

A more striking signature of the hot-phonon scenario can be detected from the analysis of the correlated lattice dynamics.

More explicitly, we focus on the mean-square *relative* displacement of atomic pairs projected onto the vector joining the atom pairs [68, 69]:

$$\sigma_{ij}^2 = \langle [(\mathbf{u}_i - \mathbf{u}_j) \cdot \hat{\mathbf{r}}_{ij}]^2 \rangle, \tag{13}$$

where $\mathbf{u}_i$ and $\mathbf{u}_j$ are the lattice displacements of atoms $i$ and $j$ from their average positions, $\hat{\mathbf{r}}_{ij}$ is the unit vector connecting atoms $i$ and $j$.

This physical quantity can be evaluated microscopically as:

$$
\sigma_{ij}^2 = \frac{\hbar}{N} \sum_{\mathbf{q},\mu} \left[ \frac{1}{2} + n\left( \frac{\omega_{\mathbf{q},\mu}}{T_{\mathbf{q},\mu}} \right) \right] \left\{ \frac{|\hat{\epsilon}_{\mathbf{q},\mu}^i \cdot \hat{\mathbf{r}}_{ij}|^2}{M_i \omega_{\mathbf{q},\mu}} + \frac{|\hat{\epsilon}_{\mathbf{q},\mu}^j \cdot \hat{\mathbf{r}}_{ij}|^2}{M_j \omega_{\mathbf{q},\mu}} \right.
$$
$$
\left. - \frac{2\mathrm{Re}\left[ (\hat{\epsilon}_{\mathbf{q},\mu}^i \cdot \hat{\mathbf{r}}_{ij})(\hat{\epsilon}_{\mathbf{q},\mu}^{j*} \cdot \hat{\mathbf{r}}_{ij})\mathrm{e}^{i\mathbf{q}\cdot\mathbf{r}_{ij}} \right]}{\omega_{\mathbf{q},\mu}\sqrt{M_i M_j}} \right\}. \tag{14}
$$

It is useful to quantify the degree of correlation by introducing a convenient dimensionless correlation factor $\rho_{ij}$ defined as

$$
\sigma_{ij}^2 = \sigma^2(i_j) + \sigma^2(j_i) - 2\sigma(i_j)\sigma(j_i)\rho_{ij}, \tag{15}
$$

where

$$
\sigma^2(i_j) = \langle [(\mathbf{u}_i \cdot \hat{\mathbf{r}}_{ij}]^2 \rangle
$$
$$
= \frac{\hbar}{N} \sum_{\mathbf{q},\mu} \left[ \frac{1}{2} + n\left( \frac{\omega_{\mathbf{q},\mu}}{T_{\mathbf{q},\mu}} \right) \right] \frac{|\hat{\epsilon}_{\mathbf{q},\mu}^i \cdot \hat{\mathbf{r}}_{ij}|^2}{M_i \omega_{\mathbf{q},\mu}}. \tag{16}
$$

The interatomic correlation factor can be thus calculated as [69]

$$
\rho_{ij} = \frac{\sigma^2(i_j) + \sigma^2(j_i) - \sigma_{ij}^2}{2\sigma(i_j)\sigma(j_i)}. \tag{17}
$$

Positive values of correlation factor $\rho_{ij} > 0$ describe a situation where the couple of atoms $i$, $j$ move in phase, so that the resulting value of $\sigma_{ij}^2$ is smaller than for the uncorrelated case. On the other hand, a predominance of counter-phase atomic vibrations is expected to result in a negative correlation factor $\rho_{ij} < 0$.

Equations (12) and (17) provide fundamental tools for investigating the effects of the hot-phonon scenario on the time evolution of the correlated lattice dynamics. At the initial thermal equilibrium at room temperature we find positive values $\rho_{\mathrm{B-B}} \approx 0.24$, $\rho_{\mathrm{B-Mg}} \approx 0.22$, reflecting a slight dominance of the acoustic (in-phase) modes in contributing to the lattice dynamics. The time dependence of the mean-square *relative* displacement for the in-plane boron-boron pair, $\sigma_{\mathrm{B-B}}$, and for the interplane boron-magnesium one $\sigma_{\mathrm{B-Mg}}$ is reported in Fig. 4a. The interatomic boron-magnesium lattice displacement (blue full line) does not show any significant feature, just a monotonic increase between the two equilibrium states, as in the absence of hot phonons (blue dashed line).

A definitive fingerprint of the crucial role of the hot-phonon is provided by the time evolution of the mean-square relative in-plane boron-boron displacements $\sigma_{\mathrm{B-B}}$ as shown in Fig. 4a. For the assumption of a thermal phonon distribution (red dashed line), $\sigma_{\mathrm{B-B}}^2$ increases monotonically obeying to a thermal heating of the lattice (similar as $\sigma_{\mathrm{B-Mg}}^2$). Within the hot-phonon scenario, however, the in-plane nearest neighbor mean-square relative lattice displacement $\sigma_{\mathrm{B-B}}^2$ presents a remarkable peak at the highest $T_{\mathrm{hot}}$ achievable at $t^* = 50\,\mathrm{fs}$, with a following decrease towards a new equilibrium conditions (red full line). Such a peak is a direct consequence of a dominance of a larger population of the in-plane boron modes with the $E_{2g}$ symmetry.

The relevance of hot-phonon physics is revealed in a even more striking way in the analysis of the lattice displacement correlation factor $\rho_{ij}$. As shown in Fig. 4b, the onset of $E_{2g}$ hot phonons in MgB$_2$ is reflected in a direct way in a remarkable anomaly in the time evolution of the relative boron-boron in-plane lattice correlation, with a sharp increase of $\rho_{\mathrm{B-B}}$ in the first 50 fs. The presence of hot-phonon modes with pure in-plane

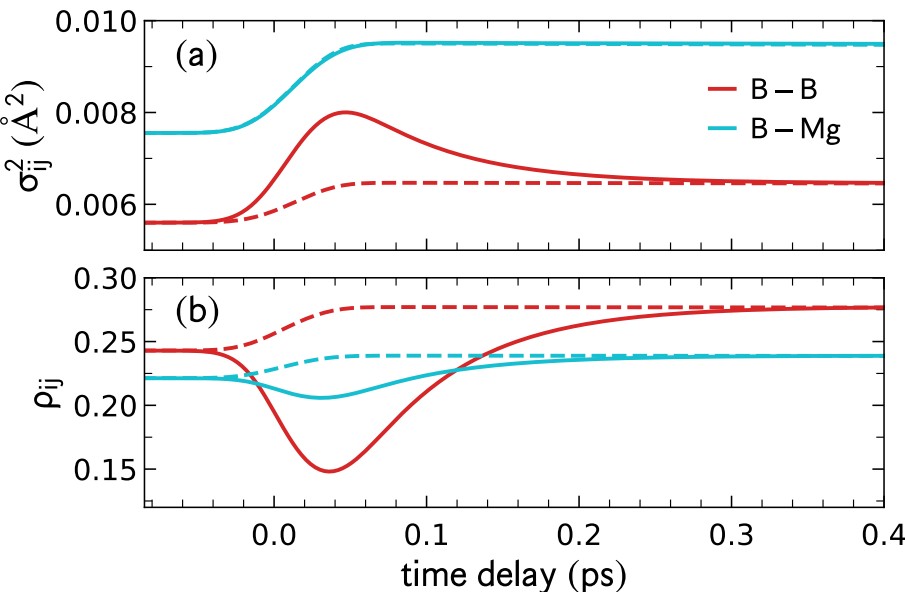

Figure 4: (a) Time evolution of the mean-square relative lattice displacement $\sigma_{ij}^2$ as defined by Eq. (14), for the in-plane nearest neighbor B-B bond, and for the nearest neighbor B-Mg bond. Solid lines show the hot-phonon scenario, whereas dashed lines represent the time behavior assuming a a thermal phonon distribution as modelled by Eqs. (9)-(10). (b) Corresponding correlation functions $\rho_{ij}$.

boron character is visible also in the interplane boron-magnesium correlation factor $\rho_{B-Mg}$, although in a weaker way. Such anomalies are peculiar of the hot-phonon scenario and are not predicted to show when the lattice modes are assumed to obey a thermal distribution (dashed lines).

Our analysis provides thus a simple and straightforward way to probe the presence of hot phonons in $MgB_2$ directly in the analysis of the lattice dynamics properties, as it can be revealed in time-resolved pump-probe experiments. More precisely, we predict a marked anomaly in the time behavior of lattice correlation factor $\rho_{B-B}$ occuring at the delay time $t^*$ when the hot-phonon lattice modes reach their largest population. The time scale over which such anomaly disappears prompts also a possible direct procedure for evaluating experimentally the time scale when *all* the lattice degrees of freedom reach their final thermalization.

## 4    Conclusion

In this paper, we have analyzed, using first-principle calculations and quantum field techniques, the presence of the hot-phonon physics in $MgB_2$ and its effect on the time evolution of lattice dynamics in a typical pump-probe experiment. We have shown that hot phonons in $MgB_2$, with their characteristic $E_{2g}$ symmetry that implies an in-plane counter-phase motion of the boron atoms, can be directly traced down in the analysis of the atom-resolved mean-square lattice displacements, and more evidently in the analysis of the mean square relative lattice displacements $\sigma_{ij}^2$ and lattice correlation factors $\rho_{ij}$. Even though we apply our investigation to the specific case of $MgB_2$, the analysis here described is quite generic and it can be extended to a wide range of materials (semiconductors, metals or semimetals) sustaining hot phonons, since they are commonly established at high-symmetry points

of the Brillouin zone, with well-defined and well-known symmetry and atomic contents. Our work provides thus a guideline for future direct experimental verification and characterization of hot-phonon mechanisms in several families of compounds.

# Acknowledgements

D.N. acknowledges financial support from the Croatian Science Foundation (Grant no. UIP-2019-04-6869) and from the European Regional Development Fund for the "Center of Excellence for Advanced Materials and Sensing Devices" (Grant No. KK.01.1.1.01.0001).

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
