# Peer review of "Fingerprints of hot-phonon physics in time-resolved correlated quantum lattice dynamics"

_SciPost Physics_

## Round 1 · Referee Report · Anonymous (Referee 1) · 2021-10-10

Report

The main aim of this work is to propose an unambiguous method to reveal the presence of hot phonons through the time-resolved lattice dynamics in Magnesium diboride.

Authors begin with the obtaining the electronic and phonon band dispersions of MgB2 and then calculate the energy transfer between different degrees of freedom which results in three different temperatures i.e., electronic temperature as well as the temperatures of the hot and cold phonons. These results are well known from authors’ previous works, e.g. in Ref. [36].

The novel part of this work is the studies of the lattice displacement correlation factor which shows an anomalous behavior when hot phonon model is employed (in compare with the simple thermal phonon model). Authors claim that such a behavior is the smoking gun fingerprint of hot-phonons in MgB2.

The manuscript is well written and the results are very interesting, however I have a few questions and concerns:

  1. Although the authors claim that their analysis is generic, but it is not obvious for me if the observed anomaly in the lattice displacement correlation factor is a unique fingerprint of the hot phonons. It seems to be just a signature of the “out-of-phase in-plane motion of the two boron atoms” in MgB2. Would it be possible to observe similar features in any system with several atoms in the unit cell, if their optical modes are excited? Or would the same measure work in other systems with different symmetries of the hot phonon modes?

  2. I think the experimental feasibility of detecting the anomalous behavior in the correlation factor should be discussed.

  • validity: high
  • significance: good
  • originality: high
  • clarity: good
  • formatting: excellent
  • grammar: excellent

Author:  Emmanuele Cappelluti  on 2022-01-05  [id 2070]

(in reply to Report 1 on 2021-10-10)
Category:
remark
answer to question
reply to objection
correction
validation or rederivation

We are greatful to the Referee 1 for his/her valuable comments and for the positive assessment of our manuscript.

Below, we provide an answer to each of his/her question.

  1. Although the authors claim that their analysis is generic, but it is not obvious for me if the observed anomaly in the lattice displacement correlation factor is a unique fingerprint of the hot phonons. It seems to be just a signature of the "out-of-phase in-plane motion of the two boron atoms" in MgB2. Would it be possible to observe similar features in any system with several atoms in the unit cell, if their optical modes are excited? Or would the same measure work in other systems with different symmetries of the hot phonon modes?

We thank the Referee for the comment. Concerning the question whether "the observed anomaly in the lattice displacement correlation factor is a unique fingerprint of the hot phonons. It seems to be just a signature of the "out-of-phase in-plane motion of the two boron atoms" in MgB2, we definitively answer: yes. "Out-of-phase in-plane motion of the two boron atoms" is an extensive way for defining the E_2g phonon branch, and the sudden dominance of these modes with respect to other ones is precisely the hot-phonon physics we are describing. We also confirm that, as stressed in the end of Sec. 1 of the manuscript, the scenario investigated in this paper that is focused on MgB2, is quite general and it can be applied to other systems, with several atoms in the unit cell and with different symmetries of the hot phonon modes. The clearest example could be underdoped cuprates [e.g., La_{2-x}Sr_xCuO_4] where hot phonons have been inferred from a three-temperature modelling of time-resolved x-ray diffraction [Mansart et al;, PRB 88, 054507 (2013)]. The symmetries of the relevant (hot) modes are shown in Fig. 12 of that paper, involving modulation of correlated motion of bond distances O-O, La-La, La-O etc. The possibility of probing the correlated motion of different pairs provides more channels for actual detections of hot phonons. Other materials (e.g., semiconducting systems such as GaAs) can be also investigated. In order to clarify in a better way this point, we have added few useful references concerning hot-phonon physics in cuprates in the end of Sec. 1 and we added a detailed discussion, and a new figure, about the perspectives in cuprates and other materials in a new Section (Sec. 4 of the revised manuscript).

  1. I think the experimental feasibility of detecting the anomalous behavior in the correlation factor should be discussed.

We thank the Referee also about this point. Considering MgB2, which is the template compound considered in this manuscript, efficient probes of the interatomic correlations has been provided in literature, at the steady level, by means of neutron diffraction, which is however not particularly suitable for femtosecond resolution. The experimental landscape is however changing day-by-day and new experimental techniques are developed, opening new promising pathways. We discuss the experimental perspectives also in the new Section 4.

---

## Round 1 · Referee Report · Anonymous (Referee 2) · 2021-11-17

Report

This manuscript is about a direct and simple method to characterize materials regarding the possible presence of hot-phonon bottlenecks slowing down the out-of-equilibrium relaxation of pumped carriers.

The proposed method avoids the technical-theoretical complications of the ultrafast, out-of-equilibrium dynamics by focusing on correlations of atomic displacements which are not explicitly time-dependent, since they are evaluated using effective temperatures in the context of the three-temperature model.

In principle, measuring this quantity may help in the experimental real-time tracking of hot-phonon behavior, as well as in theoretical investigation of candidate hot-phonon materials.

The authors demonstrate their approach on the intensely studied material MgB$_2$, where the hot phonon modes present a very specific symmetry that is easy to probe (counter-phase oscillations of the boron atoms).

I believe that the most relevant part of the manuscript starts with Eq. (13) and concerns Fig. 4 and the interatomic correlation factor.
The calculation of the boron-boron atomic correlations of MgB$_2$ using Eq. (17) was done also in Ref. [69], but at equilibrium: in that work, it was concluded that the $E_{2g}$ boron mode, though giving a large contribution, would not dominate, and therefore the correlation factor would stay positive.

In striking difference with the equilibrium case, it is found here that when the effective temperature of the $E_{2g}$ modes is raised to the hot-phonon value, the correlation factor decreases very noticeably since these densely populated phonon modes will now dominate the B-B pair-projected relative displacements with their out-of-phase oscillations. In this second case, Eq. (17) implicitly depends on time through the time-dependence of the temperature which is evolving via the three-temperature model.

I believe this manuscript to be interesting and well-written. I think it should be published in Scipost, provided that the authors reply to my points 1 to 8 listed below.

In fact, the above discussion of the interatomic correlation factor only occupies 2 pages of the 10-page manuscript, while the results of the preceding 8 pages already appeared in the literature in some form.
I do not ask the authors to shorten the text, because I think it would hurt its readability, but I would like to give some comments to better contextualize this early part.

Following point 8, I also write some additional minor points that the authors – as long as they agree – may want to address in a revised version of the text.

Requested changes

  1. About Section 2: Theoretical modelling. While I greatly appreciate the clarity of the text – everything is well-explained and easy to follow also for the non-expert reader – in my opinion this section looks like an extended retelling of part of Ref. [36] cited in the manuscript. This is a work by the same authors, among others. No new results appear then in this Section. In particular, the computational details for the DFT-DFPT calculations are the same as in Ref. [36], Figure 1(c)-(d) seems to be the same as Fig. 1 of Ref. [36], Figure 2(a) seems to be the same as Fig. 2(a) of Ref. [36], and of course Equations (1-3) – the three-temperature model – are the same as Eqs. (1-3) of Ref. [36]. If I am right and this section is mostly comprised by the same data and results of Ref. [36], I would ask the authors to write explicitly at the end of the Introduction that Sec. 2 is basically an extended explanation of part of Ref. [36], as well as to add explicitly in the captions of Figs. 1 and 2 which figures are adapted from Ref. [36].

  2. About Section 3 – Time-resolved lattice dynamics. The authors introduce here the evaluation of the (projected) mean-square lattice displacements relative to the phonon modes following Refs. [68,69]. To this end, they explain Eq. (12) and show Fig. 3. Here, the phonon statistics enters via a Bose-Einstein distribution evaluated at an effective lattice temperature: this approximates the non-thermal phonon distribution that we expect out of equilibrium. Actually, a very similar analysis was carried out in Ref. [R1] (see below), which is not cited in the manuscript. Have a look in particular at Eqs. (3), (4) and (5) of that reference, as well as Fig. 4 and the animation in their supplementary material. The only difference is that the authors of Ref. [R1] use an approximated out-of-equilibrium and time-dependent non-thermal distribution computed directly from first-principles. I would ask the authors to cite Ref. [R1], and – if they wish so – to add a sentence in order to clarify for the reader the advantages of their approach in the present context.

  3. [R1] Tong and Bernardi, Phys. Rev. Research 3, 023072 (2021)

  4. In Ref. [69], the interatomic correlation factors B-B and B-Mg are found to have a value around or below 0.1 in a temperature range from 0 to 600 K (Fig. 5c and d of that reference). In Fig. 4b of the present manuscript, the values corresponding to the same quantities are sensibly higher (0.225-0.275) both at negative time delay and at 0.4 ps, where according to Fig. 2a the temperatures should be below 500 K. Can the authors clarify this?

  5. I think the authors may be more specific about the generalization of their approach (in this, I agree with point 1. of the first Referee). In my opinion, a generalization of the proposed approach may be schematized in this way: (1) we know or suspect some specific symmetry constraints on the hot-phonon modes of some system (in this case, MgB$_2$ hot-phonon modes are characterized by out-of-phase boron planar oscillations); (2) we devise a suitable correlation factor between atomic displacements which is sensitive to such symmetries (in the present case, atomic pair projection of B-B relative displacements); (3) we check that at equilibrium / two-temperature level there is no sensible effect; (4) finally, we investigate the changes at the out-of-equilibrium / three-temperature level: if there is a marked change in correlation factor (in the MgB$_2$ case, the B-B factor drops), then we may conclude that a hot-phonon scenario is likely. Do the authors agree with this step-by-step summary?

  6. If the authors agree with the scheme in point 3: can they make another example of a correlation factor not involving out-of-phase oscillations of a specific atom but some other symmetry?

  7. I think that the authors should discuss a bit more the possible limitations of this approach. For example, the case of a correlation factor failing to “decrease”, while hot-phonon behavior is clearly observed by the three-temperature model or experimentally. Or, the correlation factor “decreasing” without hot-phonon behavior being observed or while the two- and three-temperature models give similar results.

  8. I am curious whether strong nonadiabatic effects, as in the case of MgB$_2$, would be detrimental to the quantitative accuracy of the proposed method, since the DFPT phonon normal modes become ill-defined in the presence of these effects, probably even more so out of equilibrium. What do the authors think?

  9. Also, the analysis of the interatomic correlations rests on the harmonic approximation for the phonon normal modes, and on the fact that they remain harmonic out-of-equilibrium. The authors may want to comment on this.

  10. Fig. 2(c) introduces the electron-phonon coupling strength $\lambda$ but this quantity is never defined.

  11. In Eq. (12) the phonon distribution is denoted as $n$. I just assumed that this is the Bose-Einstein distribution, which however appears as $b$ in Eqs. (5-6). Can the authors solve this discrepancy?

  12. At the bottom of page 8, the authors write “...with a sharp increase of $\rho_{B-B}$…”. I assume they mean “decrease”, considering Fig. 4(b)?

  13. Both in the abstract and in the conclusion, the authors qualify their approach as a ‘quantum-field-theory approach’. I do not understand why. I would qualify this approach as ‘semiclassical’ instead. First, the three-temperature model seems to me a semiclassical approach: Boltzmann-type equations using DFT-computed parameters (therefore at an assumed effective thermodynamic equilibrium), and in the relaxation time approximation. Second, the correlation functions between DFPT-computed atomic displacements are also expectation values on quantum harmonic oscillators in effective equilibrium, which may certainly correspond to the diagonal elements of an approximated equilibrium phonon Green’s function, but still I wouldn’t call this quantum field theory.

  • validity: high
  • significance: high
  • originality: good
  • clarity: top
  • formatting: excellent
  • grammar: excellent

Author:  Emmanuele Cappelluti  on 2022-01-05  [id 2069]

(in reply to Report 2 on 2021-11-17)
Category:
remark
answer to question
reply to objection
correction
validation or rederivation

We thank also Referee 2 for her/his careful reading and for the useful suggestions, which we have tried to incorporate in the manuscript as best as we can.

We provide below a reply for each point with the corresponding changes.

  1. I would ask the authors to write explicitly at the end of the Introduction that Sec. 2 is basically an extended explanation of part of Ref. [36], as well as to add explicitly in the captions of Figs. 1 and 2 which figures are adapted from Ref. [36].

We agree with the Referee. Sec. 2 is mainly an overview (although with some novel elements, like the direct comparison between 2T and 3T models) of former Ref. [36] (current Ref. [37] in the revised manuscript). As the Referee understand, this summary was needed and useful in order to present the basic tools for the new analysis provided in the paper and for making the manuscript self-contained. We however agree with the Referee that a most clear stress that this Section has a strong overlap with Ref. [37] is due. We have accordingly introduced a new paragraph at the end of the Introduction, with the scheme of the present work, clarifying that Sec. 2 presents theoretical modelling from Ref. [37]. We have stressed in the caption of Figs. 1 and 2 that they are a re-adaptation from Ref. [37].

  1. About Section 3 - Time-resolved lattice dynamics. The authors introduce here the evaluation of the (projected) mean-square lattice displacements relative to the phonon modes following Refs. [68,69]. To this end, they explain Eq. (12) and show Fig. 3. Here, the phonon statistics enters via a Bose-Einstein distribution evaluated at an effective lattice temperature: this approximates the non- thermal phonon distribution that we expect out of equilibrium. Actually, a very similar analysis was carried out in Ref. [R1] (see below), which is not cited in the manuscript. Have a look in particular at Eqs. (3), (4) and (5) of that reference, as well as Fig. 4 and the animation in their supplementary material. The only difference is that the authors of Ref. [R1] use an approximated out-of- equilibrium and time-dependent non-thermal distribution computed directly from first-principles. I would ask the authors to cite Ref. [R1], and - if they wish so - to add a sentence in order to clarify for the reader the advantages of their approach in the present context. [R1] Tong and Bernardi, Phys. Rev. Research 3, 023072 (2021)

We thank the Referee for poiting out to us such interesting reference that, honestly speaking, we overlooked. The approach described there, as well in other similar papers, is very promising in a compelling modelling of out-of-equilibrium time dynamics avoiding the assumption of effective temperature. In our present manuscript this level of accuracy is somehow not necessary, and, for sake of simplicity, we employ the simplest (effective-temperature based) model that captures the relevant physics. We comment on the perspectives of the approach of Ref. [R1] (presently Ref. [34]) as well as of similar papers, in a new paragraph in the Sect. 2.

  1. In Ref. [69], the interatomic correlation factors B-B and B-Mg are found to have a value around or below 0.1 in a temperature range from 0 to 600 K (Fig. 5c and d of that reference). In Fig. 4b of the present manuscript, the values corresponding to the same quantities are sensibly higher (0.225- 0.275) both at negative time delay and at 0.4 ps, where according to Fig. 2a the temperatures should be below 500 K. Can the authors clarify this?

We thank the Referee for this comment that allows us to clarify a subtle point. The Refeee is right in noticing such discrepancy in the absolute value. On this regard one should keep in mind that other scattering sources different than the electron-phonon coupling (more particularly: disorder) might play a role in the experimental analysis, affecting the quantitative estimate of the correlation functions under steady conditions. The presence of disorder is expected to lead to an overall offshifts of the value of the correlation factors, not altering the sign and the presence of hot-phonon-induced anomaly on which we focus in this paper. In order not to affect the readibility of the manuscript, we have discussed in detail these aspects in a long note (Ref. [83]).

  1. I think the authors may be more specific about the generalization of their approach (in this, I agree with point 1. of the first Referee). In my opinion, a generalization of the proposed approach may be schematized in this way: (1) we know or suspect some specific symmetry constraints on the hot-phonon modes of some system (in this case, MgB2 hot-phonon modes are characterized by out-of-phase boron planar oscillations); (2) we devise a suitable correlation factor between atomic displacements which is sensitive to such symmetries (in the present case, atomic pair projection of B-B relative displacements); (3) we check that at equilibrium / two-temperature level there is no sensible effect; (4) finally, we investigate the changes at the out-of-equilibrium / three-temperature level: if there is a marked change in correlation factor (in the MgB2 case, the B-B factor drops), then we may conclude that a hot- phonon scenario is likely. Do the authors agree with this step-by-step summary?

We confirm that the scheme of the Referee is essentially the scheme we have in mind. We thank the Referee for such feedback. Often something that appears obvious to the authors, is not to obvious for the reader. Following the suggestion of Referee 1, we have thus gladly explicitly summarized such scheme in the second paragraph of new Sect. 4.

  1. If the authors agree with the scheme in point 3: can they make another example of a correlation factor not involving out-of-phase oscillations of a specific atom but some other symmetry?

The Referee touches an interesting point. Although from the theoretical point of view, hot phonons not related to out-of-phase lattice displacements are certainly possible, from the point of view of material science the conditions to observe them might be more difficult. On the one hand, favourable conditions for hot phonons are usually realized for the zone-center optical modes, which intrinsically involve opposite displacement for inner atoms in a unit cell. However, it is still possible in principle to devise systems with many atoms in a unit cell where the out-of-phase motion of a subset of atoms is accompanied by an in-phase motion of another subset of atoms. Other possible conditions can be encountered when hot phonons are stored in the edge-zone modes. Finally, one should consider the possibility where a large electron-phonon coupling, along with possible nesting conditions, induces a strong Kohn anomaly on the acoustic branches, where also non-thermal populations can be sustained. Although such complex investigation goes beyond the aims of the present work, the search for real systems where hot phonons are associated with an in-phase lattice motion is thus an interesting subject that we hope can be further developed in the future. We have briefly commented about these aspects in Sect. 4.

  1. I think that the authors should discuss a bit more the possible limitations of this approach. For example, the case of a correlation factor failing to "decrease", while hot-phonon behavior is clearly observed by the three-temperature model or experimentally. Or, the correlation factor "decreasing" without hot-phonon behavior being observed or while the two- and three-temperature models give similar results.

We have followed the Referee's advise, smoothing some strong statements, e.g. "we suggest that a smoking gun evidence" -> "we suggest that a suitable evidence"; "it provides a striking tool" -> "it provides a useful tool". We agree that a theoretical prediction is always subject to a possible failure, because of additional and/or spurious effects, but it is hard at this point to comment about possible failures before they occur. We hope that our manuscript can provide a useful guide to experimental groups for a direct check, and we are at the moment confident that the anomalies in the correlated motion here predicted can be revealed.

  1. I am curious whether strong nonadiabatic effects, as in the case of MgB2, would be detrimental to the quantitative accuracy of the proposed method, since the DFPT phonon normal modes become ill-defined in the presence of these effects, probably even more so out of equilibrium. What do the authors think?
  2. Also, the analysis of the interatomic correlations rests on the harmonic approximation for the phonon normal modes, and on the fact that they remain harmonic out-of-equilibrium. The authors may want to comment on this.

We thank the Referee for arising these points. As the Referee might have noticed from the list of our previous publications, the role of nonadiabaticity is a crucial topic that is within the core of the main interests of the both authors. MgB2 presents an undoubtful degree of nonadiabaticity which is intrinsically entangled with anharmoncity. We therefore completely agree with the Referee that investigating the interplay between nonadiabaticity and the quantum lattice dynamics (including anharmoncity), also within the context of hot phonons, is a crucial task both for a quantitative description of the effects of the electron-phonon coupling as well as a suitable path for revealing nonadiabatic effects. This is however a formidable and delicate task where no well-defined procedure is assessed in the scientific community so that different approches have been advanced according different aims, leading sometimes to controversial debate. For these reasons, and since nonadiabaticity and anharmonicity are not fundamental ingredients of the analysis here presented, in this work we didn't address explicitly the inclusion of nonadiabatic/anharmonic effects. A proper inclusion of nonadiabatic/anharmonic effects is however a topic that we (and hopefully even other researchers) will certainly pursue in future works.

  1. Fig. 2(c) introduces the electron-phonon coupling strength lambda but this quantity is never defined.

We thank the Referee for pointing out this lack. We have added the definition of lambda in the present Eq. (1)

  1. In Eq. (12) the phonon distribution is denoted as n. I just assumed that this is the Bose-Einstein distribution, which however appears as b in Eqs. (5-6). Can the authors solve this discrepancy?

It was a typo. We thank the Referee. We have used the unique notation b(x) for the Bose-Einstein factor all through the paper.

  1. At the bottom of page 8, the authors write "... with a sharp increase of rho_{B-B} ...". I assume they mean "decrease", considering Fig. 4(b)?

We thank the Referee for having signalized us also this typo, which we have properly corrected.

  1. Both in the abstract and in the conclusion, the authors qualify their approach as a 'quantum-field-theory approach'. I do not understand why. I would qualify this approach a 'semiclassical' instead. First, the three-temperature model seems to me a semiclassical approach: Boltzmann-type equations using DFT-computed parameters (therefore at an assumed effective thermodynamic equilibrium), and in the relaxation time approximation. Second, the correlation functions between DFPT-computed atomic displacements are also expectation values on quantum harmonic oscillators in effective equilibrium, which may certainly correspond to the diagonal elements of an approximated equilibrium phonon Green's function, but still I wouldn't call this quantum field theory.

We agree with the Referee that the expression "quantum-field-theory-approach" was somehow misleading and, following the Referee's advice, we have corrected it as "semiclassical approach"

---

## Editorial Decision

resubmitted